# Sustainable Global Citizenship: A Critical Realist Approach

Jesús Granados-Sánchez 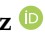

Faculty of Education and Psychology, University of Girona, 17004 Girona, Spain; jesus.granados@udg.edu

**Abstract:** The current crisis of unsustainability has renewed academic interest in sustainable global citizenship. Classical approaches to this type of citizenship have turned out to be quite abstract, utopian, and naive. This article is a theoretical reflection on sustainable global citizenship from a critical realist perspective, with the aim of bringing realism and pragmatism to the personal and social transformations necessary to achieve sustainability. The contribution of this work consists of the proposal of a conceptual framework that is structured by the following five key dimensions of citizenship: governance, status, social-ecological systems, social conscience, and engagement. These dimensions have been interpreted and described from two core ideas of critical realism: the position-practice system and the seven-scalar laminated system. The main conclusions are that agency-structure dualism requires more comprehensive approaches that integrate self-awareness of all the components that intervene in the autonomous decision to act, and that include personal capabilities, the desire and motivation to get engaged, and the real possibility of participating determined by the social context and the personal situation. It is also necessary to increase the number of types of agencies, especially with the recognition of the group as a key entity. The resolution of the dichotomy on state-global scale relationships can be articulated by differentiating between government and governance, and the role of social innovation in the latter.

**Keywords:** citizenship; sustainability; sustainable global citizenship; critical realism; agency; position-practice system; citizenship education

## 1. Introduction

Bhaskar (2002) considered that human society has entered a planetary polycrisis manifested in all four planes of social being: an ecological crisis due to our relations with nature; a crisis of morality because of how some humans treat other humans; a crisis in our social structures; and an identity crisis (who we are and whom we think we are). "*On each of these four levels we are profoundly alienated, and we are alienated by things that we cause to ourselves. They are not natural causes, they are causes which are mediated by human agency, as a result of which human beings are profoundly alienated*" (Buch-Hansen 2005, p. 63). The ecological crisis could be measured through humanity's ecological footprint (Galli et al. 2015), and it is now 73% higher than the world's ecosystems' carrying capacity. From 1 January to 29 July 2021, humanity used as much from nature as the planet could renew in the entire year (GFN & Schneider Electric 2021). Our impact is manifested as climate change, biodiversity loss, stress on freshwater, and deforestation, among many other processes and effects. Hartwig (2015) states that the roots of the planetary polycrisis are found in capitalism and that we have to move on to a post-capitalist way of doing things. For him, critical realism is above all the philosophy and social theory of transition to a planetary post-capitalist sociosphere, in which social transformation will enable universal human flourishing.

In recent years, and because of these current crises, together with the expansion and impact of globalization, there has been a renewed scholarly interest in citizenship and citizenship education with a focus on making the global agenda the citizens' agenda (UNU-CS 2018), and on assigning citizens a decisive role in the achievement of sustainable development (Granados-Sánchez 2021). If citizenship represents the normative guide to leading an

active, committed, and meaningful life, sustainability has profound implications for 21st century citizenship because it forms the founding ethical basis for rethinking the requirements, rights, and responsibilities of citizenship in a global context (Kurian et al. 2014).

Citizenship is a contested concept. Its meaning has been broadened and has become increasingly complex. The general conventional definition of citizenship includes two main meanings: membership in a political community and a form of active, responsible, and good behaviour toward the community (Gosewinkel 2010). According to the theory of citizenship (Peterson 2020), there is a distinction between different traditions of citizenship, being the liberal and the republican the main ones. In both traditions, citizenship is associated with belonging to a defined political territory and with the development of shared identity, which constitutes a crucial social link. The Liberal tradition focuses citizenship on the individual and on the rights and guarantees that the state grants and must protect and assure to the person as a citizen. On the other hand, the Republican tradition places its emphasis on the community and on the responsibility and obligations that every citizen has for the common good. For Barry (2016), the interesting and valuable thing about republican citizenship is the importance of action and contribution to the common good. Consequently, citizenship should be taught, learned, and stimulated, and the necessary conditions should be created so that it could be exercised.

Historical records tell us that when social conditions change, some aspects of citizenship issues change with them. The approach of citizenship linked to sustainable development is relatively recent and it has been theorized mainly through concepts such as environmental citizenship (Connelly 2015; Dobson and Bell 2006; Hadjichambis et al. 2020), ecological citizenship (Dobson 2006; Nash and Lewis 2006; Wolf 2007), sustainability citizenship (Horne et al. 2016), and sustainable citizenship (Barry 2006; Granados-Sánchez 2008, 2021). The sustainable development goal (SDG) 4, target 4.7 (UN 2015) states that it must be ensured that all citizens (as learners) acquire the knowledge and skills needed to promote sustainable development, a culture of peace and the practice of global citizenship (amongst other things). According to Agbedahin and Lotz-Sisitka, "*there is an absence of guidance as to how such processes can be engaged and conceptualized*" (2019, p. 104). Hadjichambis and Reis also state that "*the conceptualisation of environmental citizenship in educational context remains an imperative need*" (2020, p. 2).

This paper aims to fill this gap in the literature with a conceptualization of sustainable global citizenship from a critical realist perspective. Critical realism is a meta-theory that offers a way forward for theoretical and methodological innovation within the field of global learning (Khazem 2018). It contributes towards the understanding and achievement of transformation towards sustainability and can help to ensure the flourishing of both current and future generations. Critical realism advances the project of human emancipation and involves reclaiming reality for itself and from the ideologies that usurp, deny, and obscure it (Bhaskar and Hartwig 2011). Critical realism is concerned with other forms of explanation, the nature of causation, agency, structure, relations, and the implicit or explicit ontologies we are operating with.

The conceptualization of the link between human agency and social structures is one of the core issues in social theory (Archer 1995). Specifically, the purpose of this paper is to undertake a critical review of current interpretations of sustainable global citizenship, to broaden their scope and provide a more comprehensive, realistic and relational approach to agency and structure. The starting assumptions of this work are the following:

- Most current approaches to sustainable citizenship are based on reductionism. For example, individualism is "*a position that privileges agency over structure, considering structures as the intentional product of the activities of actors*" (Buch-Hansen and Nielsen 2020, p. 51). Humanism also places agency at the core of the explanation (Mateus and Resende 2015), while structuralism privileges structures. The agency-structure dualism needs a more holistic, integrative and relational approach (Elder-Vass 2010).
- Agency is focused primarily on an individual level of analysis (or personal agency). The conceptualization of agency should go beyond the individual to include group,

collective, and other agencies, and it should be expanded to consider all the aspects that enable or constrain each of these types of agencies in a variety of contexts.

- Agency is considered an abstract and pre-existing value based on intentionality and desire (Thorne 2005). Instead, it should be seen as a culturally informed development shaped by self-awareness of one's capacities and contextual determinations for participation in specific real practices.

This paper is a theoretical reflection on how critical realist concepts, methods and explanation models can enrich and enhance the understanding and practice of sustainable global citizenship, with a special focus on agency and structure. Its methodological processes consisted of the following three different actions and stages:

- An initial non-systematic literature review was carried out to define and conceptualise sustainable global citizenship and to identify its main dimensions.
- A second action consisted of a selection of key principles and concepts of critical realism that are useful to broaden the conception of sustainable global citizenship, in such a way that it brings realism.
- The final stage was a process of reflection on how the dimensions of sustainable global citizenship are interpreted and enriched through the lens of the principles and concepts of critical realism.

The two first processes involved a document analysis (Bowen 2009). This qualitative research methodology is a procedure for reviewing or evaluating documents. The first phase consisted of a review of the literature in the field of sustainable citizenship. Although the literature review was not a systematic one and it is not reported according to the PRISMA 2020 statement (Page et al. 2021), several documents were analysed to track change and development in the field of sustainable global citizenship and to delimit key dimensions, aspects, and appropriate questions regarding this type of citizenship. The second phase consisted of critical reading and interpretation of some works of Roy Bhaskar and other relevant authors of critical realism, with the aim of selecting those ideas and concepts of this philosophical approach that, from the point of view of the author of this work, offer powerful forms of analysis and explanation of reality.

The article is structured in three theoretical sections according to the three stages mentioned previously. The first section provides a definition of sustainable global citizenship and a selection of its main dimensions. The following section presents a brief synthesis of social realism and the description of key ideas and principles that will underpin our interpretation. In the third theoretical section, the selected key conceptual dimensions of sustainable global citizenship are interpreted following the main critical realist principles and concepts. The article ends with a conclusion section where the main arguments of this theoretical reflection are briefly summarized, and their educational implications are discussed.

## 2. Sustainable Global Citizenship

Sustainability citizenship is unclear and controversial (Nelson 2016). Currently, some authors use the concepts of environmental, ecological, and sustainable citizenship as equivalents to escape the debate of differentiation of terminologies and they choose one of them as an umbrella concept that includes them all at once. This is the case of the European Network for Environmental Citizenship (ENEC 2018; Hadjichambis et al. 2020), which considers that, from an educational point of view, it is important that there is a single concept and that the most appropriate one is that of environmental citizenship. For their part, Van Poeck and Vandenabeele (2013) choose sustainable citizenship as a uniting concept. In my opinion, the umbrella concept is used to avoid the terminological debate, but it is important to emphasize that conceptualization is not banal or neutral, it is intentional, and the nuance is important.

Environmental citizenship has placed its emphasis on claiming environmental rights, that is, the right that all human beings have to a healthy and adequate environment to develop a life with health and well-being, and this requires society and governments to

protect and ensure a quality environment. For Barry (2006) it is a very narrow and limited concept of citizenship, since it focuses mainly on environmental aspects and forgets key dimensions of sustainability such as society, politics, culture and the economy. In addition, it contains a rather passive liberal conception of citizenship and is mainly circumscribed at local and state levels. Kurian et al. (2014) see environmental citizenship as a continuation of the status quo, as citizens carry out symbolic actions such as planting a tree or consuming in an environmentally friendly way. Neoliberal environmental citizenship theorists shifted the focus from politics to economics, replacing the citizen as a political being with the consumer (an economic being), the state with the corporation, and politics with markets (Cao 2015). In this sense, sustainability depends on consumer choice.

Ecological citizenship, as described by Dobson (2003, 2011), differs from environmental citizenship in four essential characteristics:

- Citizens' obligations and responsibilities are understood as being non-reciprocal and must come first. The main obligation of each citizen is to ensure that their individual impact does not impair the possibilities of others to have opportunities and to meet their needs. Dobson (2004) uses the ecological footprint as a measure of this impact.
- Virtue is a very important concept in citizenship. Justice and equity stand out as first-rate virtues. Justice has to be applied spatially and temporally, that is, among all citizens of today and of the future and regardless of their origins (intragenerational and intergenerational justice). A second order of virtues includes care and compassion.
- Both the public and private spheres of people are considered since private individual and group acts often have implications for the public sphere. The private sphere is closely linked to standards and lifestyles.
- Ecological citizenship extends beyond the state because it needs to be global (it must include the whole Earth). For Wolf (2007) it is a citizenship without territoriality since national borders are nothing more than an obstacle to effective action on problems such as climate change, which is caused by, affects, and has to be remedied with the participation of all terrestrial citizens, governments, and global actors. Thus, ecological citizenship needs to be global by definition.

These four characteristics are the basis of what Dobson calls a post-cosmopolitan citizenship. The main criticisms of Dobson's theorizing about ecological citizenship highlight the fact that the author insists a great deal on the individual agency of citizens and little on other types of agencies, in addition to ignoring the social, economic, political, and cultural structures that restrict people's ability to act (Sáiz 2005).

Barry (2006) prefers to talk about sustainable citizenship, as he considers it more ambitious, multifaceted, and challenging because it cares more about social justice, equity, and democratic governance. Sustainable citizenship is well aware of the structural causes of socio-environmental degradation and, above all, deepens the social and economic aspects of sustainable development, such as respect for human rights, social inclusion, otherness, solidarity, equality and equity, quality of life, deliberative democracy and participation for good governance. Bullen and Whitehead (2005) argue that the notion of sustainable citizenship destabilizes the spatial, temporal, and material parameters on which forms of modern citizenship are based. The norms of sustainable citizenship go beyond state boundaries and the present to include individuals who do not know each other, either because they live in distant parts of the world or because they are people who are yet to be born. It is not reciprocal and assumes that individuals do not act on the basis of personal gain or by claiming certain rights, but because they feel responsible for others and are motivated to act to achieve social justice and equity (Atkinson 2014). For Van Poeck and Vandenabeele (2013) a sustainable citizen is an active, critical and independent citizen who is able and willing to play an active role in solving problems and issues related to sustainable development. Dobson (2011) also conceives sustainable citizenship as a pro-sustainability behaviour, both in the public and private spheres, based on equity in the distribution of environmental goods and on the co-creation of policies for sustainability. Precisely, Kurian et al. (2014) affirm that deliberative processes constitute the key element

of sustainable citizenship because they go beyond mere routine dialogues and allow the confrontation of ideas that can bring out shared values and carry out sustainable actions that combat, for example, inequality as a result of political and economic power. Barry (2016) goes one step further and proposes that citizens have the obligation to carry out a service to the sustainability of the community, in the spirit of contributing to the common good.

In this paper, the author uses sustainable global citizenship because the concepts of sustainability and sustainable development are more holistic and intentional than that of the environment and, moreover, direct us towards a horizon. I, therefore, believe that it is more appropriate as an object of citizenship. In addition, using sustainable global citizenship makes it easier to link the field of citizenship with that of education for sustainability, two fields that until now have evolved in parallel and without a clear and determined connection that must be solved. Last, but not least, the ascription of the global scale to citizenship is important for highlighting the importance of our role in the current and future health of our planet.

There is no widely agreed definition of global citizenship. As stated by UNESCO (2017, p. 2), "global citizenship does not entail a legal status. It refers more to a sense of belonging to the global community and a common sense of humanity, with it presumed members experiencing solidarity and collective identity among themselves and collective responsibility at the global level". Political approaches to sustainable global citizenship include globalist (Beck 2010) and pluralist theories (Cao 2015) to emphasize the importance of interconnection and interdependence, universalism, plurality, and the need for inclusion of diversity, and difference. Abdi (2015) advocates decolonising global citizenship by problematizing the basic meanings and assumptions coming from Western countries and moving to an epistemic pluralism. For him, "the current mono-epistemicalizations of global citizenship education which are disempowering and de-culturing people in more ways that we can count here, should be redesigned and reconstructed with multi-locational knowledge and cultural pluralisms that can effectively and inclusively respond to the realities of lived citizenship contexts that are not fixed or static but are active and dynamically shifting as demanded by the contexts and relational categories that sustain them" (Abdi 2015, p. 23).

As Nelson (2016) argues, despite the fact that there is no consensus when it comes to defining sustainable citizenship, we can identify the main characteristics that make it distinct from other forms of citizenship. Table 1 contains three synthetic lists with the main dimensions that could structure the approach to sustainable citizenship, according to the opinion and analysis reported by the selected authors. Most of these dimensions are represented by concepts that constitute dualisms or dichotomies.

**Table 1.** Examples of key dimensions of sustainable citizenship.

| Dobson (2010) | Kurian et al. (2014) | Nelson (2016) |
|---|---|---|
| Territorial and Non-territorial Citizenship | State and non-state | Pragmatic Glocal Citizenship |
| Rights and Obligations | Rights and Responsibilities | Collective Responsibilities and Obligations |
| Active and Passive Citizenship | Democracy and Capitalism | Participatory Democracy and Shared Governance |
| Public and Private spheres | Public and Private | Socioecological approach |
| Individual and Community | Universal and Particular | Individual, People and Environment (one, the other and otherness) |
| Virtue | Human and Non-human | "Being" rather than "Having" |

Synthesis of the works of Dobson (2010); Kurian et al. (2014) and Nelson (2016).

The first dimension of sustainable citizenship confronts state-based sovereignty and how it is linked to political territories, with other non-territorial forms of citizenship. Secondly, the three cited authors coincide in pointing out the importance of rights, responsibilities, and obligations. Regarding this, Nelson (2016) suggests going beyond the individual scale and emphasizes the importance of the collective in relation to responsibilities, obligations, and rights. In third place, the authors address citizen participation in decision-making from different angles: Dobson (2010) pays attention to the active or passive attitude of citizens, while Kurian et al. (2014) and Nelson (2016) focus on the political system, the latter stressing the need for participatory democracy and shared governance in order to achieve sustainability. Although agency and structure play a decisive role in this dimension, the authors do not emphasize these two elements beyond political participation. The next dimension differentiates the public and private spheres, or how to overcome this dichotomy with a more inclusive approach. Another characteristic of sustainable citizenship highlighted by these three authors is the opposition to individualistic and collective positions. In their analysis, they do not consider the group as a category that is decisive, as well as the diversity of complex relationships that occur between individuals, groups, and the collectivity. The last key dimension for Dobson (2010) is virtue. The sustainable citizen does the right thing, not because of the incentives, but because it is the right thing to do (Dobson 2003). People often choose to do good for reasons other than fear of punishment or loss, or a desire for financial reward or social status. People sometimes do good because they want to be virtuous (Beckman 2001). According to Dobson (2010), justice is the primary virtue, with empathy, compassion, and caring being secondary virtues. Nelson (2016) stresses the need to understand self-realization outside of materialism and to move from "having" to "being".

### 3. Critical Realism Key Principles and Concepts Related to Citizenship

Critical Realism is a philosophical approach to the functioning of society proposed by Roy Bhaskar (1978). As a meta-theory, critical realism is concerned with the investigation, discussion, and structure of other theories to improve our ability to understand the world and, therefore, reality. To Bhaskar (1998) there is a world independent of our knowledge about it and this world both pre-exists as a condition and is reproduced or transformed by human action.

According to Archer et al. (2016) defining critical realism is not an easy task because there is not a unitary framework and social realist scholars may have a heterogeneous series of positions, but the feature which unites them is a commitment to the following critical realist principles:

- *Ontological realism*: the world is real, structured, and complex. It exists independently of our knowledge or awareness of it.
- *Epistemological relativism*: knowledge is a continuous process that is socially produced under specific social, historical, and cultural conditions and is, therefore, changeable, and fallible. As knowledge is contextual, conceptual, and activity dependent it must embrace a form of epistemic relativism.
- *Judgemental rationality*: Bhaskar and Hartwig (2016) state that, despite the fact that knowledge is fallible, it is possible to arrive at decisions between competing beliefs of theories because not all interpretations are epistemically or morally equal and there can be rational grounds for preferring one to another (or judging which accounts about the world are better or worse).
- *Cautious ethical naturalism*: this principle is an attempt to reconnect facts and values. Facts are value-laden, and values are fact-laden and, therefore, values are open to empirical investigation and critique.

These principles relate to three domains of reality (or strata of knowledge): the real, the actual and the empirical (Mateus and Resende 2015). The real domain refers to whatever exists, be it natural or social ("*everything there is*"). The real cannot be observed and exists independent from human perceptions, theories, and constructions. It includes the power-

generating structures and the event-generating mechanisms. The actual domain refers to what happens when these powers and mechanisms are activated and produce change (“*everything we can grasp*”). The empirical domain is what we know about reality and that we gain through experience and with the perception of the effects of actualities (“*everything we can observe*”). Therefore, critical realism makes a clear distinction between the real world and the observable world and represents a shift from epistemology to ontology and from events to the mechanisms that produce them (Agbedahin and Lotz-Sisitka 2019).

The *Transformational model of social activity* (TMSA) (Bhaskar 1998) is a key theoretical contribution that makes a distinction between individuals acting and the society that enables and constrains their actions. For Bhaskar (1998) the relationship between structure and action is transformational and not dialectical because they don't constitute two moments of the same process. Societies are the material condition and the result of human action and, therefore, it is both the ever-present condition and the continually reproduced outcome or result of human agency. On the other hand, praxis is the reproduction or transformation of society. The TMSA offers two central ideas for the conceptualization of sustainable global citizenship: the *position-practice system* and the *seven-scalar laminated system*.

The TMSA suggests that agency emerges from the structure via the *position-practice system*, a point of contact or mediating system between human agency and social structures. This mediating system is key to understanding the evolving, dynamic and unpredictable open system of reality. To better understand the concept of the position-practice system, Bhaskar (1998) points out that positions should be understood in a more broad and elaborate way, as the positions that a person occupies and the practices in which is engaged because of these positions. The post or position involves functions, rights, duties and tasks and certain degrees of assumption and enactment. The practices are the activities within the system in which individuals are involved. For this reason, the position-practice binomial is conceived relationally and as part of a system. States, corporations, intergovernmental organizations, NGOs, and other group and collective partnerships also have an agency and a position-practice system as entities in social systems and structures and global governance.

The seven-scalar laminated system is another core idea of the TMSA that is useful for analysing and examining emergent and complex issues. Table 2 shows the seven scales or levels and their descriptions according to their potential in agential transformative praxis.

**Table 2.** The seven-scalar laminated system.

| Levels and Scales | Description |
| --- | --- |
| **Level 1.** The sub-individual psychological scale. | It is concerned with the intrinsic personality of the individual. It includes the individual's nature, identity, character, and psychology, as well as the individual's motivation, aptitude, confidence, intentions, interests, desires, and concerns. |
| **Level 2.** The individual person or biographical scale. | Describes the person that is studied and their capacity to determine the impact on social events and actions. The individual agency and the position-practice system are influenced by the state of mind and body of individuals (such as being healthy or sick, being capable and skilled, having access to training and resources, and so on). |
| **Level 3.** Small group micro scale. | The micro level represents the studied group of the population, especially when individuals interact. Human interaction denotes the relationship between the individual, the group, and the collective, and can develop an emergence of group and collective agency as a result. |
| **Level 4.** The meso scale: structures and functional roles. | This level of the laminated system explores structural factors that give rise to individual and collective experiences (focusing, for example, on relations between functional roles). Structural factors include mechanisms and/or powers which may be known or unknown, constructive, or destructive, and pleasant or unpleasant. |
| **Level 5.** The macro scale: societies and territories. | This layer of reality is concerned with the functioning and operation of societies and/or their territories/regions. The understanding of societies as a whole includes knowledge about its composition, constitution and configuration, and how these elements influence individuals, groups, collectives, and structures. |
| **Level 6.** The mega scale. | It is the analysis of civilizations and their traditions, which are the result of different geo-historical trajectories. |
| **Level 7.** The planetary scale (or cosmological level). | It is the superior level, and it understands the planet as a whole (as a planetary system). Global bodies and agencies such as the United Nations have a key role in developing, implementing, and monitoring global policies. |

Author's synthesis from the following sources: Bhaskar (2010); Price (2014) and Agbedahin and Lotz-Sisitka (2019).

## 4. Sustainable Global Citizenship Key Dimensions through the Lenses of Critical Realism

Dobson (2010) and Kurian et al. (2014) argue that sustainable citizenship requires a deliberative dialectic on dichotomies, that is, it deals with deliberation based on the discussion of pairs of opposing or contradictory ideas and the confrontation of these in a variety of contexts so that shared values are found. Dialectics allows for interpretative flexibility and achieves synthesis, poses options and aids decision-making. A critical realist notion of sustainable global citizenship goes beyond reductionist and deconstructionist approaches to dichotomies and dualisms because reality is complex. "*Critical realism constitutes an ambitious attempt to transcend all the dichotomies/dualisms (...) by offering a nuanced both/and perspective as opposed to an uncompromising either/or perspective.*" (Buch-Hansen and Nielsen 2020, p. 49). Table 3 presents the key dimensions that, from my personal interpretation of critical realism and sustainable global citizenship, should structure the conceptualization of this type of citizenship. These dimensions are governance, status, social-ecological systems, social consciences, and engagement. The dimensions are described in the following sections of the paper, and they are characterized by unique concepts that represent citizenship's main areas of attention. In turn, each dimension contains a series of threshold concepts arranged in the form of dualisms, but which also integrate different nuances and possibilities. The intention behind this approach is to focus on the relationships, interactions, and dynamics between the two poles of each dualism, as well as the relationships between dualisms in the same dimension and with other dualisms in other dimensions, with a special focus on agency and structure.

**Table 3.** Key dimensions and threshold concepts of sustainable global citizenship, from a critical realist interpretation.

| Dimensions | Threshold Concepts and Dichotomies | |
| --- | --- | --- |
| Governance | Sovereignty, state sovereignty, national governments, legality, power, statal authority. Territoriality. | Global governance, international relations, multi-level politics, cosmopolitanism, planetarity, diffusion of authority. Non-territoriality, aterritoriality. |
| Status | Responsibilities, duties, obligations. Belonging, membership, identity. | Rights, guarantees. Exclusion, multiple identities. |
| Social-ecological systems | Individual. Personal. One. Public. Society, social processes. | Group(s), communities, collective/collectivity. The other, otherness. Private. Nature, ecosystems, natural processes. |
| Social conscience | Agency ("agencies"). Self-consciousness, self-efficacy, locus of control. | Structures (social, political, economic), culture. Unconsciousness. Social innovation. |
| Engagement | Commitment. Participation, empowerment, action, activism. Self-determination, codetermination. | Inhibition. Inaction, passivity. Hesitance. |

Author's own proposal.

### 4.1. Governance

The concept of citizenship has a long history in political science, but in recent decades, it has been reinterpreted in many ways and has acquired multiple meanings beyond the political. One of the current main challenges in citizenship is how we combine our agency related to the state and the agency beyond the state and sovereignty.

Agnew (2018) repudiates the false dichotomy of state sovereignty and the global as antithetical political realities. I also disagree with the conventional vision in which it is believed that globalization and states must compete with each other. On the contrary, my conception is of complementarity, which leads to an approach of dual reality, with the first

component that is national citizenship (which maintains the political sovereignty of citizens circumscribed in the nation-state and their role in the election and control of a government), and a second component that is global citizenship understood as an alternative construction of "the global" and globalization, from governance and social innovation. Government is a formally centralized political authority that administers a territory or state and executes policies in a hierarchical relationship and within a legality. On the other hand, governance is characterised by a diffusion of authority and power, where non-state actors, such as transnational corporations and social movements, engage in processes of decision-making (Rüland and Carrapatoso 2022).

In the following subsections, the paper presents a critical realistic vision of citizenship that integrates its political, social, cultural, and economic variables, in situations of territorial sovereignty (what is known as a nation-state), in global governance (through multi-level politics in different territorial scales and situations), as well as in other non-territorial life spheres. The dualism state-global could be analysed through the seven-scalar laminated system model. The state and its government are represented by the meso and macro scales (levels 4 and 5), while the global constitutes the planetary scale (level 7). Individual and group agencies of citizens (levels 2 and 3) are affected by the subindividual psychological scale (level 1), and it is of a different nature depending on the levels where action is taken or affected. In this sense, as we move vertically or horizontally within levels, we pass from state to global realities, and from sovereignty to social innovation.

### 4.1.1. Citizenship and Territorial Sovereignty

Traditionally, citizenship has been related to belonging to a geographically delimited community with which we identify and share an identity, a political space, sovereignty, and a legality. Nation-states are the territories that have played a decisive role in the determination of citizenship because it is where we have the greatest capacity for political decisions or sovereignty in all its scales, from local to state. Sovereignty is a question of the real status of people, which is why the legal system of the country from which we originate applies to us and from which we benefit from civil, political, and social rights, as well as taking on obligations and responsibilities. This conception of citizenship is one of the fundamental components of the socio-legal framework of the Westphalian world of state political communities (or modern states) (Cortés and Piedrahita 2011). For Dalton (2008), citizenship in democratic states is characterized by a series of rules established and regulated by the community itself and which determine what is understood as a good citizen. This expectation revolves around four basic principles:

- Public participation is necessary to make sense of and legitimize the democratic process.
- Autonomy presupposes that citizens develop their opinions separately from each other, through information and dialogue, enabling them to understand various points of view and to form their own.
- Citizens accept the legitimacy of the state and of obey the law.
- The relationship with others means that citizenry includes ethical and moral responsibility towards others based on justice and solidarity.

At present, citizenship as an idea and as a practice is multidimensional and its link to a territory or country is today insufficient (Anderson 2008) for several reasons. First, it does not respond to the complex realities of states. Second, as Dobson (2010) points out, territoriality is a discriminatory characteristic of citizenship since it is a condition or requirement of belonging that, when fulfilled, means privilege and when not fulfilled, such citizenship is denied and, therefore, it excludes. Third, states alone do not provide solutions to the needs and challenges imposed by globalization, climate change and other major problems of global unsustainability are included in the 2030 Agenda (UN 2015). Fourth, there are sovereignties of an integrationist type (Agnew 2018), such as the European Union, which transcend the nation-state and are causing the centrality of states to diminish in favour

of territorially augmented sovereignties, with a recognition of supra-state citizenships (Malatesta and Granados-Sánchez 2013).

Although sustainability transcends the borders of countries, I do not believe that the idea of sovereign states should be discarded as proposed by Kurian et al. (2014) because it is not realistic. Instead, states should be considered as a reality with which we live, and which can be an object of transformation or susceptible to being transformed through collective agency. Nor do I believe that states should have such a central and coercive position as Barry (2016) advocates.

4.1.2. Global Citizenship

The Anthropocene suggests to us that we belong to a unique human community that, with its form of development, is influencing the destiny of ourselves and the planet. This notion of human community has led to various conceptualizations such as global citizenship (UNESCO 2018), planetary citizenship (Gadotti 2017), and terrestrial citizenship (Morin 2004), to mention just a few. For Morin, "*we humans have a common identity: not only the same genetic code, the same brain capacity, but the same capacities of emotion, of sympathy, of friendship and, therefore, of hatred. Likewise, among us we have a community of destiny* " (Morin 2004, p. 73). This common destiny also underlies the approach of Gadotti (2017), which supports planetary citizenship in a unifying vision of the planet and of a single-world society that practices "planetarity" and considers the planet as an intelligent evolving being. For the author, planetary citizenship is, in essence, active, full, and just in social, political, cultural, institutional, and economic terms and implies a planetary democracy.

According to Delanty (2015), cosmopolitanism is a normative idea about the world that is located in the broader totality or world community (kosmopolis) and that gives relevance to the perspective of the other and to compassion for the rest of humanity. Diogenes is attributed to be the first to pronounce himself a citizen of the world (in Greek kosmospolitês) and to originate cosmopolitan thought in classical Greece. However, cosmopolitanism has been developed as a theory throughout history: in his essay on perpetual peace, Kant (1795) called for the creation of a global federation of states that would enter into a cosmopolitan order where all people would be treated as equals, forming a universal community. Kantian cosmopolitanism recognized an ethic of hospitality that would diminish the meaning of state borders and that should abandon the idea of hostility between countries and people. Cosmopolitan citizenship also promotes inclusion, impartiality, and non-discrimination (Bullen and Whitehead 2005). It is fundamentally different from nation-state-based citizenship because it does not involve making distinctions about who is a citizen and who is not and who is inside and who is left out. Cosmopolitanism seeks global dialogue and allows everyone to have a voice and speak from their perspective, although in the end, the best argument is the one that should prevail.

In recent decades, globalization has posed a challenge to the world order for many reasons: economic activities have expanded beyond state borders; networks and trade flows have increased and intensified; interactions and the dissemination of information and knowledge have accelerated; people's mobility has increased and expanded; and all these global processes are increasing the impact on the planet and have altered the functioning of states (Held 2002). Globalization is an asymmetrical and stratified process because not all citizens of the world are in the same position to "be global" and, as a result, the benefits generated by globalization are unevenly distributed (Dobson 2010) and benefit some elites and a small group of people. Or put another way, not all of us influence and contribute to the polycrisis mentioned earlier in the same way. We all can be affected by the consequences of globalization, and by the risks posed by climate change, but "*there are sustainability issues and risks that are peculiar and more prevalent in some continents and their respective member states than in others*" (Agbedahin and Lotz-Sisitka 2019, p. 108). For example, Africa has a relatively small carbon footprint compared to other continents in the world but is particularly affected by the effects of climate change and other unsustainable problems highlighted by the Agenda 2030 (UN 2015), such as hunger, food insecurity,

malnutrition and poverty. In short, individuals, groups, states, and regions influence and are affected by global processes unevenly. At the same time, "*we need to appreciate the complex lived contexts of people*" (Abdi 2015, p. 20). Thus, there is a principle of differential responsibility towards global issues that must be assumed by individuals, groups, and states, depending on their global positions over time.

Cortina (2004) proposes not leaving globalization to chance and orienting it towards voluntarily desired goals. To that end, she believed that international institutions must be reformed and new ones created to ensure transnational communities that join through agreements. According to Fraser (2007), this hypothetical new transnational public power should allow and guarantee the legitimacy of public opinion through the communication process that includes all those affected, regardless of their political citizenship. These contributions of political cosmopolitanism and global citizenship are being criticized for a lack of realism and because they arise from a Western, elitist perspective that focuses on globalization (which has a negative connotation for the rest of the world for its social and ecological consequences) and on the construction of utopian global political systems that do not exist today and that are more than questionable (Boni and Calabuig 2015). So, it would seem more appropriate and realistic to flee from the pretension of global political organizations with their unknown consequences and which compete with states.

Dingwerth and Pattberg (2022) identify two cross-cutting themes that global governance needs to engage: non-state agency and the complexity of global governance structures. The first feature of global governance is that we need to talk about governance in the global rather than governance of the global. There is a multicentric world made up of constellations of diverse actors and governed by hybrid institutional complexes. Transnational non-state actors are key in global governance and include intergovernmental organizations, international non-governmental organizations, multinational corporations, social movements, transnational professional communities, and globally influential individuals. Some of these actors play a role in governance through orchestration or through influence. Some pursue a lucrative interest while others do not. The second key issue in global governance is the risk of overemphasizing the global scale and the Western vision. Global governance is a multi-level system that includes political processes and governance systems at all levels of human activity, from the family to the international organization, and all are inseparably linked. Scale interrelations (both vertical and horizontal) should be considered as part of global governance, and there is an overgeneralization of Western experiences that hinders the visibility and development of the agency of the South.

It seems that there is a move towards another type of political activity that is already materializing and being channelled through social movements and transnational networks (Kartal 2012), which additionally have the function of acting as a counterpower. Global civil society is uniting for causes related to human rights, environmental conservation and many great causes that affect humanity as a whole and are carrying out collective actions that are generating changes. This new global citizenship nears the idea that Dobson (2011) called post-cosmopolitanism, which contemplates both relations between citizens and citizen-state relations and posits that citizens' obligations have a socioecological nature and must be guided by virtue. Post-cosmopolitism focuses on developing the sense of interconnection and interconnectivity to feel that we are part of multiple collectives and groups at different territorial or aterritorial scales. In addition, taking account of interconnection involves looking at the present and the future, as well as the implications and consequences of the actions carried out at any given time.

Finally, it is important to bear in mind that global citizenship can also be understood through life facets different and away from politics. This way of conceiving and living life as a common vital project (Gimeno 2003) could not be attached to a specific territory and be based on certain values that bring together people from different backgrounds (and without universalist pretensions) who share the same way of seeing the world and a horizon to which they tend. The Internet facilitates digital environments where individuals can create communities and exercise new types of citizenship. Digital citizenship as described by

Ribble (2015) lacks elements of digital action and community building and engagement. These elements are critical for the development of what Elder-Vass (2022) calls digital utopias (utopian projects that emerge in the digital space).

*4.2. Status*

Human beings belong to political communities that confer them citizenship status that grants benefits, as well as ascribes duties. Just as countries have different political realities, the status of citizenship and its characteristics will also differ from country to country. The classic discourses of citizenship have focused on the recognition of the rights, responsibilities and obligations that exist within the functioning of the political structure of a community or a state, which arise as a result of social relations. Rights have evolved and expanded over time but with spatial variations and singularities. Marshall (1992) distinguished an initial threefold typology of rights (civil, political, and social) that has been increased up to four in recent years (Granados-Sánchez 2008). The first set of rights achieved were civil rights, including the right to freedom of expression and the right to free association. Political rights came after and, thanks to them, most citizens in many countries obtained the right to vote. Later social rights linked to the welfare state were won and, finally, environmental rights are recognised today in many countries. The tensions in the field of rights are related to whether these should be considered socially or privately based, whether they should be a combination of individual rights and community rights, and how to seek universality and attention to diversity with specific rights for minority groups with particular needs.

The dimensions of responsibility include taking into account the object and subject of responsibility, as well as the territorial scales concerned. As objects of responsibility, we find oneself, the "we," other members of the global community, and future generations. There must also be responsibility for the Earth and all forms of life. Individuals, groups and collectives are the subjects to whom responsibility is assigned. Therefore, there will be individual, group and collective responsibilities and duties. According to Wolf (2007), responsibility is related to the sense of belonging and the perception of interconnection with others that can drive us to the obligation to act. Thus, responsibility arises from the idea that one is part of local, regional, national, and global communities, and other horizontal groupings from which one has or perceives interconnection and to which it associates values of solidarity, security, and sustainability.

The sustainable post-cosmopolitan citizenship of Dobson (2010) is based on the principle of global dialogue and socio-ecological obligations. The obligation goes beyond the immediate reality to extend it to the global, and to lives lived in other parts of the planet and in present and future times. Obligations do not belong equally to all humanity, since the impact that each produce is different. Therefore, the obligations have to be asymmetrical depending on the position and impact of each citizen in the global sphere, in different moments. Dobson (2004) focuses on reducing the ecological footprint, but our obligations must go much further and extend to social, political, and moral aspects. Asymmetric obligations also have to be extended to groups, corporations and nations based on the inherited world order and the impact on the global change generated (such as, for example, the contribution to climate change), highlighting diluted communities and dense communities with historical obligations.

In the context of sustainability, both rights and responsibilities are important, because achieving sustainability is a collective undertaking. Deliberation between rights and obligations requires negotiating ideas of sufficiency, sustaining nature, and fulfilling obligations to present and future generations. Ultimately, the responsibility of sustainable global citizenship is to ensure efficient governance at all levels, from the local, regional, state, supra-state, and global levels.

### 4.3. Social-Ecological Systems

Nature is an essential part of our lives that must be incorporated into the discourse and exercise of sustainable global citizenship. The concept of social-ecological systems refers to the relationships between nature and society, and it is used to describe, analyse, and model human-nature interactions. The social-ecological systems are defined as hybrid and emergent systems resulting from interactions of various social and natural components over space and time (Liehr et al. 2017). The key components of social-ecological systems are ecosystem functions and social actors. Ecosystem functions are part of natural dynamics, but they can change due to societal actions. Thus, they can provide services and goods, or disservices and harm society. Social actors include individuals and groups of persons who influence ecosystem functions with the practice of their agency. Görg et al. (2017) find that *"what is largely missing in the current transformation debate are analyses that focus in more depth on the interactions between globalized societies and the natural environment, analysing resource use patterns and its social implications in terms of global inequalities as much as its impact on global ecosystems without denying local (including everyday), regional and national scales of problems and action"* (2017, p. 5).

One claim of this article is to include the group as a key category in citizenship and, thus, transcend the individual-collective dualism. Schild (2016) also advocates for the need for an expanded view on citizenship that goes beyond individual-level analysis and the personal duty and lifestyle approach. Wight (2006) points out that not only individuals have emergent properties. The activities of groups of people can also display emergent properties that cannot be reduced to the activities of single individuals. Among these emergencies is the development of identities linked to belonging to various groups, as well as the influence that these close groups exert for the development of a type of agency different from the individual. Hill (2012) argues for self-organized, action-oriented, problem-solving groups enabling people to recognize structures of social control and power relationships, develop oppositional forms of action, generate novel ideas and create new kinds of knowledge that can help us in a better relationship with nature. In sum, both in nature and in society, there are individuals (and phenomena) that can act alone or united in groups or collectivities, and these can be governed by certain particularities. In turn, these groups are subject to laws of universal value. In this way, the individual, the particular, the group, the collectivity and the universal are mutually linked. According to Görg et al. (2017) *"a critical concept of social-ecological transformations points at a better understanding of the social-ecological dimensions of current transformation processes. This includes a better understanding of scale interactions, i.e., global, regional, and local processes, and the systemic processes as much as the actor constellations and power relations involved"* (2017, p. 6).

Another tension in the field of sustainable global citizenship is the different points of view between individualised and profit-seeking self-interest promoted by current hegemonic neoliberalism and the global common good. In other words, the dualism between private and public interests. From liberal political theory, the public sphere is the space of politics, power, and civic engagement. The private sphere, on the other hand, is the intimate family space of care and protection, where all personal things are protected from the power of the state. Republicanism places public interest before private interest, which implies distinguishing that something that may be good for an individual may not be good for the same individual as a member of a community (Dobson 2006). Inherent to this is the explicit commitment to freedom as non-domination and the acceptance of plurality in the way of living. Sustainable global citizenship recognizes that the public and private spheres, although different, are interrelated since private acts can have public implications and vice versa, and both have an impact on natural systems and on the achievement of sustainability. According to Dewey (1954) democracy (and sustainability) must begin at home and has to be built on face-to-face interactions in which human beings work together cooperatively to solve the ongoing problems of life. Small groups and bigger partnerships are the places where relationships and trust are built. These elements, together with the

transdisciplinary and transformative competence (Granados-Sánchez 2022), are essential for effective group and collective engagement.

One cannot separate and oppose the public from the private, the individual from the group and the collective, or personal life from political life. To Biesta "*the most important transformation that is at stake in the experiment of democracy is the transformation of private troubles into public issues*" (2015, p. 5), but I would also add that we need to transform the public troubles into private concerns. He continues pointing out that "*what is always at stake in the democratic experiment is the question to what extent and in what form private "wants" (that what is desired by individuals or groups), can be supported as collective needs (that is considered desirable at the level of the collective), given the plurality of individual wants and always limited resources*" (Biesta 2015, p. 5). This leads us to reflect on how to satisfy our needs and desires in simple ways and move from "having" and "pretence and appearance" to "being".

### 4.4. Social Conscience

Bhaskar states that "*in order to explain what a person does, you must make reference to their reasons, their conscious intentionality*" (in Buch-Hansen 2005, p. 60). Conscience is like a person's moral and emotional compass that provides a sense of what is right and wrong. "*Individual conscience compels us to act morally in our daily lives ( . . . ) whereas social conscience compels us to insist on moral action from the wider institutions of society and to seek the transformation of social structures that cause suffering ( . . . ) most people experience a gap between the kind of world they see and the kind they want. On a personal level, social conscience is what bridges that gap. If we understand our own social conscience, we can make more conscious choices to help shape society according to our values*" (Goldberg 2009, p. 105).

Bhaskar's (1998) critical realism provides us with good tools for the development of a more accurate social conscience. This theory maintains that reality is formed by different layers and by internal elements united in a multiplicity of complex structures and interrelationships. The ontological principle of critical realism is precisely the stratification of social reality. Social consciousness has to allow us to know reality and our role in it at any moment. Social consciousness is the knowledge that a person possesses and that is imbued with value judgments about things. According to Goldberg (2009), social conscience is composed of three interrelated elements: agency, structure, and consciousness. Agency is defined as the ability of individuals to decide and act independently or in agreement with others and implies a sense of free will, choice, or autonomy (Hay 2002). It is the sense of personal power and constitutes personal responsibility. However, as Hill (2012) points out, human agency not only depends on the wishes of people; to get involved in the community, in addition to the will, it is necessary to have the skill and the opportunity; that is, it is about seeing where it is possible to intervene personally and with others, where we have room for decision and action, and if we are able to do so. Thus, agency is conditioned cognitively, by emotional and affective aspects, character traits, contextual frame of mind, self-efficacy, and performativity. There are disagreements among critical realists as to what aspects of agency are most prominent (Buch-Hansen and Nielsen 2020). Some authors recognise the importance of the unconscious and habits (following Bourdieu), whilst others ascribe great importance to reflexivity and intention of agents, and the interactions between individuals. In addition, sustainable citizenship should emphasize the importance of "agencies", because actions are not always just individual: there are group and collective actions that involve a larger number of people and the community. It should also be remembered that many groups and minorities, due to their particularities and fit and treatment in society, do not have the same capacities and opportunities to engage in political, social, and civic activities and are systematically set apart or excluded (Jones and Gaventa 2002).

Social structures are abstract entities of economic, political, social, and juridical-legal origin that organize the life of the community. Social structures are context-dependent, and they are human activity-dependent in time and space. These structures facilitate and motivate agency and make it possible and, at the same time, limit or constrain it. That

is why it is so important to know the structures, their construction and reconstruction mechanisms, in order to participate in public life.

Buch-Hansen (2022) raises the question of to what extent human beings have agency. His reply to it, from a realist perspective, is that humans are always confronted by structures, yet exercise agency. "*While those structures constrain actions, they never determine them. Human beings are understood to have their own unique capabilities that are irreducible to, but not unaffected by, social structures and culture. Such capabilities vary from person to person: for instance, a human being can be more or less strategic, rational, creative, kind*, etc. *On this view, then, human beings are able to exercise agency*" (Buch-Hansen 2022, pp. 10–11).

For critical realism, the relationship between structure and agency is not dialectical but rather recursive and transformational, since society (structures) and human practice (agency) do not constitute moments of the same process: society and its structures are the existing material condition for praxis (which is later) to take place. Agency comes after the structure: "*there was structure; there is now that agency; and there will be the structure that this agency produces*" (Buch-Hansen 2005, p. 63). The result of praxis can reproduce social structures, or it can transform them and lead us to a new reality. Thus, society is both an ever-present condition and a result of continuous transformation. Archer (2017) has advanced in the comprehension of the agency-structure relation through the concept of morphogenesis. Her model focuses on how the interplay between structure and agency can be studied by cycles of structural conditions, social interaction, and structural development. It builds on the notions of stratification and emergence so that structures and agency have emergent properties of different natures. Archer (2010) also separates culture from structures because culture concerns existing relations between ideas. The confrontation of complementary or conflicting ideas takes place in sociocultural interactions, constituting another ontological dimension. In sum, the activities of agents are conditioned both by social structures and a cultural system.

Structures create a system with hierarchies, and people and groups occupy different places in this reality, stratified according to the moment and context. In addition, the resources and facilities of the system are available differentially among social actors. As a result, power and the possibility of participation and action are unevenly distributed among individuals, groups, and institutions. Having social conscience allows us to know the position we occupy in the system and the power and possibilities that this gives us to be able to act and transform reality. The dialectic of critical realism states that as people engage personally and with others, they can generate new ideas and transform structures through collective agency (transformative praxis) (Khazem 2018).

Bhaskar (1998) understands emancipation as being the transition from an unwanted, unnecessary, and oppressive situation to a desired, necessary, and empowering situation that allows people to flourish. Social innovation is a process that allows social emancipation. For Belda Miquel et al. (2019, p. 27) "*social innovation from citizenship for the transition to more sustainable models is a process of innovation through collective action with multiple dimensions: it is a process of individual and collective transformation in the micro, but also a process of experimentation, testing and prefiguration of possible systemic transformations on another scale; it is a learning process, and it is a complex and contradictory process. This innovation does not put the State, but the citizenry, at the centre, but it does so by challenging public policies. It constitutes spaces for mobilization that concretize and make the demands of the neighbourhoods and the general indignation against the system real, since they already draw and practice other possible models.*" Social innovation is the social interaction scenario that allows collective agency.

### 4.5. Engagement

Civic engagement is one of the leitmotifs of most republican approaches to sustainable citizenship and it is manifested through commitment, participation, and action (and their opposites).

Participation is the force that guides and legitimizes democracy. Banks (2008) classifies citizenship according to the level of participation of citizens into the following four types:

- The most superficial level of citizenship is *legal citizenship*, which applies to citizens that don't participate in politics in a meaningful way.
- *Minimal citizenship* refers to legal citizens that just vote in local and state elections.
- *Active citizenship* involves actions that are designed to support and maintain existing social and political structures.
- *Transformative citizenship* requires the promotion of social justice through the execution of actions that challenges and changes current norms and structures.

This classification is useful for understanding the degree of our commitment to sustainability, from inhibition to transformation. On the other hand, it has some limitations: it is focused on the individual, and it presupposes that a citizen always participates in the same way and degree in all aspects of his/her life, while commitment and engagement depend on the context. Some authors such as Macedo et al. (2005) and Putnam (2000) maintain that the political participation of young people, in most Western societies, is problematic because it is in serious decline. Hibbing and Theiss-Morse (2002) affirm that there is an apathy towards politics and the public and that citizens do not want to be involved in political decisions or know the details that influence and determine decision-making. If we look at participation in the last Spanish elections in 2019 and the American elections in 2020 and 2022, we can see that political participation is revitalized if the context demands it and the electorate is mobilized. For Dalton (2008), the conclusions about the decline in political participation are not entirely correct since they have considered only part of the political activity ("institutionalized" political participation) as if they were the only rules of citizenship. According to him, there is a multitude of rules of citizenship and forms of political participation and action that are simply changing (some are weakening, while others are strengthening). Thus, while citizen obligations such as voting in elections could be eroding in certain periods, other ways of engaging in political action are increasing. Inglehart and Welzel (2005) believe that we are at a time when new generations in advanced countries with better education have changed political party membership by actively participating in specific social movements or interest groups, which allow for direct action and greater influence on outcomes, although this requires greater dedication and commitment. This coincides with the results of research on citizenship and climate change carried out by Wolf (2007), in which the participants stated that they seek to act and develop their agency on issues that are important to them beyond governance and the top-down opportunities that the state provides them. In short, we are facing a renaissance of democratic participation (Dalton 2008) understood in a very personal way and motivated by desire for a self-realization of the individual and groups through having a higher quality life and social relationships. As a result, we are facing a challenge to the traditional political elites, moving from the "politics of loyalties" (or of supporting political institutions and parties) to the "politics of election" (Atkinson 2014), where citizen participation is configured through collective social actions separate from the state (Jones and Gaventa 2002).

One of the main objectives of education for sustainability since its inception has been to provide the appropriate tools for each citizen to develop action competence so that he or she can participate actively and meaningfully in public life and in the private sphere. Some scholars go further and foster activism and even transgression (Lotz-Sisitka et al. 2015). Should we all be activists all the time, or should we be self-conscious critical citizens who engage differentially, considering personal and social necessities and complexities? There are no real emancipatory proposals that train citizens in agency self-awareness and self-regulation of their own individual, group, and collective participation. Our foundation is based on the ideas of Goldberg (2009), who proposes that there are many causes, problems, phenomena, or objects of interest, while the time available to us is limited. *"Individuals cannot care deeply and act effectively on every social and ecological problem they come across, but they can identify problems they feel are important and that they have the agency to act on"* (Goldberg 2009, p. 108). This implies that we first need to decide if we want to act or stay passive and inhibited in each situation. Inaction may be due to health issues or other reasons of a personal nature that require our attention and energy and prevent us from

engaging. If we choose to get involved in actions, we need to know which causes we join, know with whom we join, what barriers or structural possibilities we face, what real possibilities of intervention exist and decide our level of involvement and commitment (Granados-Sánchez 2011, 2021). These decisions will not only depend on the social urgency, because our personal context at that moment will also be decisive, which will depend on our training and knowledge of the phenomenon to be treated, along with our motivation, personal values, state of mind, the position we occupy in the group or collective to which we integrate, and the time available. Our sense of self-efficacy and locus of control are also decisive. Self-efficacy is the belief in one's own abilities to succeed in a particular situation. Locus of control is the degree of control a person thinks he or she has over what happens in their life, as opposed to outside forces.

Participatory democracy needs horizontal and bottom-up forms of shared governance, as well as top-down opportunities for civic participation. Thus, governments and public institutions as well as social movements and global networks have to provide opportunities for citizens to participate in the elaboration of policies or actions that encourage sustainability. It is also about the citizenry being able to create spaces, programs and even structures in which social groups and actors can visualize and materialize sustainability from carrying out daily sustainable practices. It is about executing, creating transformation processes, and executing them as a result of collective effort. It is living and breathing the bond with the community (Nelson 2016). It is about creating social learning and building human and social capital.

## 5. Conclusions and Educational Implications

This paper approached sustainable global citizenship from a critical realist perspective. The position-practice system and the seven-scalar laminated system are two core concepts of critical realism that have helped in the analysis of the selected key dimensions in the field of sustainable global citizenship and how they relate to agency-structure dualism. The main conclusions of this theoretical reflection are the following:

- *Governance*: The practice of citizenship takes place in communities to which we belong and are linked. These communities may have a territorial component and may or may not be linked to sovereignty and legality or be non-territorial or aterritorial (Carpenter 2019). At present, the legal recognition of the sovereignty of the citizens of the world is limiting in terms of territorial scope. In any case, the nation-state does not determine the entire space of participation or the feeling of social, cultural, or other belonging. The challenge of sustainable global citizenship is, precisely, to recognize that there are other territories as well as communities on which we have a vital and moral link beyond sovereignty, and to determine how we articulate that reality outside the nation-state and how we participate in a political, social, cultural, moral and economic way on a global scale, but also in different parts of the world, because the global citizen is a citizen involved and affected by all spatial scales, the territorial interrelationships, and the non-territorial communities to which he or she joins by other facets and objectives that they share. In short, we need to integrate engagement in both government and governance through participation in politics and in social innovation initiatives, to monitor and control policies and to act as counterpower.

- *Status*: global citizenship is stratified and asymmetrical. There is a principle of differential responsibility towards the global that must be assumed by individuals, groups, and states, according to their impact due to the global positions they occupy over time. At the same time, big efforts should be made to achieve symmetry in the status of all people.

- *Social-ecological systems*: Sustainable global citizenship is based on the principle of socio-ecological obligations and global dialogue. Nature and its processes should be integrated in sustainable global citizenship as essential elements and not as external entities. Or put another way, we should consider ourselves as part of the social ecological systems, both as individuals and as part of groups or collectives. The

group is a category that needs more attention because many of our actions occur in connection with close people and this relationship of familiarity produces emergencies that strongly determine our agency as individuals and beyond. Social-ecological systems have distinctive characteristics and threats according to scale and context.

- *Social Conscience*: To explain reality and social outcomes, we have to consider the ways in which agency, structure, and culture are intertwined. This approach is more comprehensive and goes beyond other reductionist proposals that give centrality to the agency and, specifically, to abstract active participation. Social conscience enables citizens to have real power over their agency.

- *Engagement*: There are many factors that determine civic engagement and participation, and citizens should be aware of them in order to decide how to act as autonomous and conscious individuals at all times and thus avoid social apathy and frustration.

This theoretical reflection is an initial stage in the development of the conceptualization of sustainable global citizenship from a critical realist perspective. Its limitations include the lack of detailed specification or development of some of the key components of agency (and their relationships with the structure), that have been described throughout the analysis of the six dimensions.

We could say that there is a general agreement in understanding education for sustainable citizenship as a process that has to do with the acquisition of knowledge, skills, competencies, and values that allow sustainable development, understood as a sign of good citizenship. However, the way in which we understand how this should be carried out is not so clear. To Hadjichambis and Reis (2020) the different political and philosophical approaches make the fostering of citizenship related to the environment and sustainability very complex for educational practice. The latest trends in sustainable citizenship and global citizenship increasingly point towards transformative learning (UNESCO 2018). Mezirow (1995) focuses transformative learning on the cognition of individuals but does not propose how cognitive transformation leads to social action or the development of the individual, group, and collective agencies, especially for the collective transformation of human actions. According to Lotz-Sisitka et al. (2015), all forms of transgressive and transformative learning require pedagogical methodologies that involve a multitude of voices and social actors and that make co-learning, cognitive justice and the development of individual, group, and collective agencies possible. For Biesta (2011), it is about moving from a socializing learning focused on the learning of norms, to a learning based on experiencing democracy and sustainable development. That is, learning to develop sustainably by carrying out sustainable actions that, in addition, make democracy extend, grow, and even lead us to different democratic forms that are more relevant. Education therefore becomes not merely a key method of achieving sustainable development, but part of the development itself. This type of education involves citizenship as practice (Lawy and Biesta 2006) or citizenship in practice (Van Poeck and Vandenabeele 2013). What Biesta (2011) calls "democratic experiment" constitutes a process of transformation of the person and the community and involves exceeding the preconceived standards for sustainable global citizenship and moving from thinking about what skills citizens should acquire to promoting sustainable spaces and practices. Thus, education emerges as a space in which citizens are invited to explore issues that concern and occupy them (Van Poeck and Vandenabeele 2013).

In conclusion, we could say that education for sustainable global citizenship from a critical realist perspective should be based on respect for difference and diversity, relational knowledge, the importance of consensus and disagreement, critical self-reflection, the ability to navigate between the general and the particular, individual and collective responsibility, and the ability to decide how to get involved, act and commit oneself (Kahn and Agnew 2015). It involves the advance in planetary awareness and the culture of sustainability (Gadotti 2017). Future research on education for sustainable global citizenship should focus on the development of multiple identities (Sen 2006), both as citizens of a nation-state, as global citizens, as employees, as consumers and as members of many territo-

rial and non-territorial communities. It should entail the experience of sustainability, which challenges people to manage the complexity of reality at all scales, and which requires the understanding of and working with social actors because sustainability is the result of a collective effort. Sustainable global citizenship should make us reflect on our place in an increasingly connected and network-intensive world, on ourselves and the other, us and them, here and there and that, in the end, we are all one.

**Funding:** This research received no external funding.

**Institutional Review Board Statement:** Not applicable.

**Conflicts of Interest:** The author declares no conflict of interest.

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
