# Peer review of "Sustainable Global Citizenship: A Critical Realist Approach"

_socsci, doi:10.3390/socsci12030171_

Round 1
Reviewer 1 Report
Dear authors, the subject of the paper is really interesting, but come corrections should be done to increase the overall quality of the paper.
1. In the abstract the contributions to the current literature should be mentioned, also the applicability of the result should be specified.
2. The aim of the paper should be supported by references, which will show the gap in the literature, that you are trying to fill in.
3. Methodology part should be improved.
- In the first stage the methods of " The selection of key critical realist principles and concepts that are related to citizenship" should be specified.
- In the second stage the selection criteria and the whole process of a literature review should be explained in detail.
- The third stage of methods "A reflection on how the dimensions of sustainable global citizenship are interpreted through and enriched with the lenses on the critical realist principles and concepts" is not the part of analysis, it is the discussion part, and it should be removed from the methodology section.
4. Please check carefully the section 4. Discussion. As its content is not relevant to the paper.
5. Conclusion part should be expanded, it should include main results of the analysis, limitations of the study, theoretical and practical implementations of the results, recommendations for further studies.
Author Response
Dear reviewer,
Thank you very much for your comments and suggestions. I really appreciate them. They have helped me in the improvement of the paper.
Next, I will explain the changes that I have introduced in each section. I hope the new version of the manuscript meets your requirements.
Dear authors, the subject of the paper is really interesting, but come corrections should be done to increase the overall quality of the paper.
- Thank you very much for considering of your interest the theme of the paper.
- In the abstract the contributions to the current literature should be mentioned, also the applicability of the result should be specified.
- A new abstract has been written. It includes the aim of the article, some findings and key ideas from the conclusions.
- The aim of the paper should be supported by references, which will show the gap in the literature, that you are trying to fill in.
- The introduction has been considerably expanded. From line 70 to line 76 there is a justification of the need for a conceptualization of sustainable citizenship with to references from the literature. The following paragraph explains the aim of the paper (lines 77-86). And the text continues with some starting assumptions that are also found in the literature (as requested) (see lines 87-106).
- Methodology part should be improved.
- The final part of the introduction explains the methodology of the paper. This is a theory-oriented article. My purpose is to present a personal theoretical reflection on sustainable global citizenship from a critical realist perspective.
- In the first stage the methods of " The selection of key critical realist principles and concepts that are related to citizenship" should be specified.
- Section 3 of the article is dedicated to critical realism key principles and concepts related to citizenship. This section introduces critical realism as a philosophical approach and justifies the two core ideas/concepts that are chosen.
- In the second stage the selection criteria and the whole process of a literature review should be explained in detail.
- A new paragraph in the introduction has been added (lines 119-129) in which the literature review process is briefly explained. This process is documented and described further in the second section: “Sustainable global citizenship”. From lines 237 to 268 there is an explanation why three proposals have been found and used to help in the creation of a personal contribution for the conceptualization of sustainable global citizenship.
- The third stage of methods "A reflection on how the dimensions of sustainable global citizenship are interpreted through and enriched with the lenses on the critical realist principles and concepts" is not the part of analysis, it is the discussion part, and it should be removed from the methodology section.
- I have followed your directions and the paper now presents a new section as a discussion part. This is section 4 that is titled: Sustainable global citizenship key dimensions through the lenses of critical realism. The contents of this section have been considerably extended.
- Please check carefully the section 4. Discussion. As its content is not relevant to the paper.
- My apologies. The file of the manuscript that I first submitted was not the right final version. I uploaded a previous version that still had some unfinished parts. In this new version of the manuscript, I have removed that content.
- Conclusion part should be expanded, it should include main results of the analysis, limitations of the study, theoretical and practical implementations of the results, recommendations for further studies.
- The conclusion part has been expanded as requested. Lines 815-864 are new content that includes a summary of the main results (815-858), the limitations (860-864), the educational implications (865-892) and some future directions (899-907).
I hope the new version of the paper has captured all your comments.
My best wishes.
The author.
Reviewer 2 Report
GENERAL REMARKS AND SUGGESTIONS
Overall, this is a well-written and well-structured paper with acceptable methodological justifications. The analyses and arguments put forward attest to the authors’ conversance with the relevant scholarship. Beyond pinpointing the gaps in the literature on citizenship and sustainability, the authors made viable suggestions for a more successful education for sustainable global citizenship. The authors’ critical engagement with key concepts, the flow of arguments, the mastery of relevant aspects of critical realism and the rich concluding reflections, make the paper look promising. All the same, a few minor issues need to be addressed to ameliorate it.
SPECIFIC COMMENTS AND SUGGESTIONS
· ABSTRACT
The authors should refer to the concluding reflections (5. Conclusions and final reflections: educational implications) in the Abstract.
· INTRODUCTION
The introduction should clearly provide readers with the main axes around which the paper revolves. Although the authors alluded to these points in the subsections ‘Aim of the paper’ and ‘Methodology’, making them conspicuous would be appropriate.
· DISCUSSION (PAGE 10)
I don’t know how useful this section is to the paper. I suggest the authors rid the paper of this section.
Author Response
Dear reviewer,
Thank you very much for your comments and suggestions. I really appreciate them. They have helped me in the improvement of the paper.
Next, I will explain the changes that I have introduced in each section. I hope the new version of the manuscript meets your requirements.
GENERAL REMARKS AND SUGGESTIONS
Overall, this is a well-written and well-structured paper with acceptable methodological justifications. The analyses and arguments put forward attest to the authors’ conversance with the relevant scholarship. Beyond pinpointing the gaps in the literature on citizenship and sustainability, the authors made viable suggestions for a more successful education for sustainable global citizenship. The authors’ critical engagement with key concepts, the flow of arguments, the mastery of relevant aspects of critical realism and the rich concluding reflections, make the paper look promising. All the same, a few minor issues need to be addressed to ameliorate it.
- Thank you very much for your kind comments and I am glad to hear that the paper looks promising.
SPECIFIC COMMENTS AND SUGGESTIONS
- ABSTRACT
The authors should refer to the concluding reflections (5. Conclusions and final reflections: educational implications) in the Abstract.
- A new abstract has been written. It includes the aim of the article, some findings, and key ideas from the conclusions.
- INTRODUCTION
The introduction should clearly provide readers with the main axes around which the paper revolves. Although the authors alluded to these points in the subsections ‘Aim of the paper’ and ‘Methodology’, making them conspicuous would be appropriate.
- The introduction has been considerably expanded. From line 70 to line 76 there is a justification of the need for a conceptualization of sustainable citizenship with to references from the literature. The following paragraph explains the aim of the paper (lines 77-86). And the text continues with some starting assumptions that are also found in the literature (as requested) (see lines 87-106). The final part of the introduction explains the structure of the paper as suggested (see lines 130-137).
- DISCUSSION (PAGE 10)
I don’t know how useful this section is to the paper. I suggest the authors rid the paper of this section.
- My apologies. The file of the manuscript that I first submitted was not the right final version. I uploaded a previous version that still had some unfinished parts. In this new version of the manuscript, I have removed that content.
The paper has been considerably extended. Section 2 has included an explanation about how the key dimensions in sustainable global citizenship have been identified. Section 3 of the article is dedicated to critical realism key principles and concepts related to citizenship. This section introduces critical realism as a philosophical approach and justifies the two core ideas/concepts that are chosen. Section 4 that is titled: Sustainable global citizenship key dimensions through the lenses of critical realism. The contents of this section have been improved and extended. The conclusion part has included a summary of the main results, the limitations, the educational implications and some future directions.
I hope the new version of the paper has captured all your comments.
My best wishes.
The author.
Round 2
Reviewer 1 Report
Dear author, thanks for your revised paper.